# Testing a Model of Human Spatial Navigation Attitudes towards Global Navigation Satellite Systems

**DOI:** 10.3390/s22093470

**Published:** 2022-05-03

**Authors:** Carmen Moret-Tatay, Maddalena Boccia, Alice Teghil, Cecilia Guariglia

**Affiliations:** 1MEB Lab, Faculty of Psychology, Universidad Católica de Valencia San Vicente Mártir, Burjassot, 46100 Valencia, Spain; mariacarmen.moret@ucv.es; 2Department of Psychology, Sapienza University of Rome, 00185 Rome, Italy; maddalena.boccia@uniroma1.it (M.B.); alice.teghil@uniroma1.it (A.T.); 3Cognitive and Motor Rehabilitation and Neuroimaging Unit, IRCCS Fondazione Santa Lucia, 00179 Rome, Italy

**Keywords:** global navigation satellite system, attitudes, gender differences, aging

## Abstract

Global navigation satellite systems (GNSS) can provide better data quality for different purposes; however, some age groups might lie outside its use. Understanding the barriers to its adoption is of interest in different fields. This work aims at developing a measurement instrument of the adoption attitudes towards this technology and examining the relationship of variables such as age and gender. A UTAUT model was tested on 350 participants. The main results can be summarised as follows: (i) the proposed GNSS scale on human spatial navigation attitudes towards geopositioning technology showed optimal psychometric properties; (ii) although statistically significant differences were found in the Wayfinding Questionnaire (WQ) between men and women, these did not reach the level of statistical significance for the scores on attitudes towards GNSS; (iii) by testing a model on human spatial navigation attitudes towards geopositioning technology, it was possible to show a higher relationship with age in women.

## 1. Introduction

Moving from one place to another is essential for full citizenship in both new and familiar environments. Those who struggle with this process require clinical [1] and/or external support (such as other people or technological assistance) [2]. In this way, technology adoption is a complex process where different variables might interfere, such as gender [3], age [4], and even attitudes towards technology adoption [5]. In this context, the digital era has offered an emerging market action, where some items that were considered economically unachievable decades ago are currently available [6]. One example that might reflect the role of digitalization in current society is mobile digital wayfinding. Nowadays, by using GNSSs (Global Navigation Satellite Systems), we are somewhat less likely to look up to or create a mental map to make a journey, avoiding the inherent cognitive effort to do so. In this way, a route is suggested by the GNSS, avoiding other options to be considered in most cases. By letting the device set an available route, networks in our brain do not need to retrieve possible paths and select the best option, and other cognitive functions are not employed to achieve this goal (such as executive functions, memory, and attention) [7,8,9].

The literature has shown that structural changes occur within specific areas of the brain throughout the aging process, which can negatively affect human navigation [10]. These changes include a decrease in the frontal structures of the brain and the entire striatum (typically affecting egocentric strategies) as well as a reduction in the volume of the hippocampus (associated with allocentric strategies and general spatial learning). However, these differences in navigation are more pronounced due to lesions in the brain in people with mild cognitive impairment and dementia; indeed, topographic disorientation is one of the most common issues in Alzheimer’s disease (AD) and is considered one of the first symptoms of the disease. Regarding the role of GNSSs in the elderly, a study employing functional magnetic resonance imaging (fMRI) scans shed light on differences between non-dependent and dependent GPS (Global Positioning System) users in older adults [11]. The former showed higher activity and a higher volume of grey matter in the hippocampus than those depending on GPS.

Some literature also suggests that the regular use of GNSS not only has an effect on the formation of a map of the environment but also on wayfinding processes that involve the planning and choice of routes based on previous knowledge and environmental cues [10]. In this context, human spatial navigation can be described by two specific cognitive processes to deal with distances: egocentric and allocentric strategies. The first uses distances and indications to or from individual reference points with respect to the individual’s body position. In turn, allocentric navigation is a world-centered strategy that uses information about distances and angles between different locations in the environment, regardless of the individual’s position [8]. It should be noted that egocentric navigation has several similarities with GNSS instructions, as both of them reduce the load of cognitive demand to a specific target, such as “turn right”. Moreover, the literature has claimed that depending on GNSS to navigate may have a harmful effect on brain areas such as the hippocampus, which is involved in memory and navigation [11,12]. In brief, the brain activity generated in the hippocampus and prefrontal cortex of participants circulating the streets of a London neighborhood through computer simulation was higher when the number of options to choose was increased, but no additional brain activity was detected when people followed GNSS instructions. Nevertheless, for a cognitive map to be useful, not only the use of these strategies but also different cognitive sub-processes may occur, such as the estimation of distances or even feeling processes of fear and unease, such as anxiety.

Even if GNSS technology has been adopted for decades, some age groups, such as older adults who lie outside its use, are of interest in this field [13,14]. According to the authors, the adoption of this type of technology would be closely related to other non-technological aspects of everyday spatiotemporally, as well as to the design of technologies that do not correspond to the lower perceptions of the cognitive abilities of older adults. Other variables, such as technology anxiety and lower self-efficacy among the older population [4,12,15], could be responsible for the heterogeneity of results. The literature has shown high individual differences in estimating distances [3,4]. Gender differences are a matter of debate in the field, as numerous studies found an advantage for men in navigation in real and virtual environments, as well a description of higher use of orientation strategies in men rather than women [16,17]. Indeed, in line with the evidence from a self-report instrument, males have been reported to choose shortcuts more often rather than females, who tend to follow known routes [18,19]. Additionally, considering gender differences in technology adoption [13], it is more than likely that women will not adopt digital technology to geoposition themselves in the environment. This can be considered paradoxical, as this technology aims to support geospatial awareness. However, a cohort effect might occur here, as these differences seem to disappear for visuo-spatial abilities and spatial navigation when participants are trained in spatial navigation, such as in the case of pilots [20].

Understanding spatial-navigation attitudes are of interest in the field of technology adoption, as their use enhances full citizenship. Many models have been proposed in recent years to help explain the acceptance or adoption of technology. Among these, the Unified Theory of Acceptance and Use Model (UTAUT) by Venkatesh [21] was chosen as the theoretical framework for the present study. Accordingly, four core variables to determine a user’s behavioral intention (BI) to use an item can be described as follows: performance expectancy (PE), effort expectancy (EE), social influence (SI), and facilitating conditions (FC). These variables can be related to most navigation complaints and adapted to the UTAUT model, as depicted in Figure 1. It should be noted that one of the most widely used instruments in this field is the Wayfinding Questionnaire (WQ), a self-report questionnaire to assess navigation complaints that might affect most of the UTAUT variables of interest [22].

The use of questionnaires has helped not only to understand the issues underlying the intention to use such devices but also to collect information on visuospatial navigation complaints such as anxiety, as described in the WQ subfactor. However, to our knowledge, the literature on attitudes towards the adoption of GNSS across the lifespan, as well as other underlying variables to spatial navigation, is rather limited. The novelty of this work is to provide a tool for the assessment of attitudes towards GNSS and to study its role within a technology adoption model. For this purpose, the current research had three main goals: (i) to develop a questionnaire on attitudes towards global navigation satellite systems; (ii) to examine the proposed questionnaire on attitudes towards geopositioning technology with the Wayfinding Questionnaire (WQ); (iii) to examine gender and aging effects by testing a model on attitudes towards global navigation satellite systems.

## 2. Materials and Methods

### 2.1. Participants

A sample of 350 participants (196 women; 56%) volunteered in the study. The mean age was 36.36 (*SD* = 13.46), with a range of 18 to 65 years. Most of the participants were single (36.6%), whereas 30.2 % were married or living with their partner, 2.3% were widowed, 5.2% were divorced, and 25.6% were in a romantic relationship. A total of 66.3% of the participants were actively working, while 23.3% were full-time students, 6.4% were unemployed, 2.3% were retired, and 1.7% were housekeepers. Lastly, 96.5% were right-handed.

### 2.2. Procedure

A cross-sectional study was carried out. Anonymous data were collected using an online questionnaire shared via social media from a university on south-eastern coast of Spain. Participants were informed about the procedures and aims of the study. They were also reassured about their right to withdraw from the study at any time, the voluntary nature of their participation, and that all data provided by them would be treated as confidential. Informed consent form was obtained from all participants, and ethical approval was obtained by the University Ethics Committee (UCV/2019-2020/122).

### 2.3. Materials

The main measures employed were two specific questionnaires described as follows:(i).A questionnaire on attitudes towards geopositioning technology was employed (GNSS scale). The proposed questionnaire was developed through three different stages. The first stage, called the preliminary stage, consisted of developing and assessing the content of the items. In order to do so, a group of experts with proven experience in both formal and informal education was selected. A list of 5 items was obtained that contained several proposals for the concept described as one factor. After experts rated the items from 1 (minimum) to 10 (maximum), items with a value lower than 8 were not included. The last version contained 4 items, which were included in the final analysis, as described in Appendix A. The internal consistency for the proposed questionnaire on adoption of global navigation satellite systems was optimal, with an ω = 0.823.(ii).The Wayfinding Questionnaire (WQ) [23,24], in its Spanish adaptation [25], comprises 3 subscales: navigation and orientation, distance estimation, and spatial anxiety. In terms of internal consistency, WQ was described as follows: ω1 = 0.915 for navigation and orientation (Navigation), ω1 = 0.858 for distance estimation (Estimation), and ω1 = 0.789 for Spatial Anxiety (Anxiety).

### 2.4. Data Analysis

Data analysis was performed using SPSS (IBM, Armonk, NY, USA) and its AMOS plug-in, JASP (Version 0.12.2, Amsterdam, The Netherlands), and SmartPLS3.3.3 (Oststeinbek, Germany). Descriptive analysis was carried out; normality and homogeneity were tested as well. The internal consistency was examined with McDonald’s ω. A confirmatory factor analysis (CFA), accompanied by goodness-of-fit indices, was conducted. Confirmation of the adequacy of the model used absolute fit indices; chi-square statistic X^2^; comparative fit index (CFI) with a reference value of 0.90; the normed fit index (NFI), also called delta 1; incremental fit index (IFI), with a reference value similar to CFI; error of the root mean square approximation (RMSEA), for which the smaller the value, the better the fit, where the reference value is 0.06. A non-parametric approach was adopted to address differences across gender. As data normality was not achieved, a partial least-squares regression (PLS) was also carried out. According to literature, the benefits of this approach are discussed in terms of sample size and normality assumptions [26]. The Mean Absolute Percentage Error (MAPE), Mean Absolute Error (MAE), Mean Square Regression (MSR), Q^2^, and R^2^ were chosen to measure different error metrics.

## 3. Results

First, a CFA on the questionnaire on attitudes towards global navigation satellite systems (GNSS) was carried out. An optimal goodness of fit was found: χ^2^/_degrees of freedom_ = 4.19; NFI = 0.983; IFI = 0.987; CFI = 0.98; RMSEA = 0.09. Once the factor structure of the new questionnaire was confirmed, descriptive analysis across gender was carried out. Scores on the GNSS scale were slightly higher for men (mean = 23.18, *SD* = 4.25) rather than for women (mean = 22.68, *SD* = 5.48). As expected, these did not reach statistical significance. Differences between WQ subfactors across gender reached the statistical level, with higher scores for men in Navigation (mean = 5.58, *SD* = 0.94 versus mean = 4.88, *SD* = 1.07) and Estimation (mean = 4.89, *SD* = 1.43 versus mean = 3.73, *SD* = 1.44). However, in terms of Anxiety scores, men reported lower scores than women (mean = 3.36, *SD* = 0.83 versus mean = 4.01, *SD* = 1.02). As shown in Table 1, a non-parametric approach was chosen, as the variables studied did not reach equality of variance across groups (Levene test with *p* < 0.05).

A correlation analysis was also conducted on the variables of interest, including Age, and was conditioned by gender. Table 2 and Figure 2 show correlations under Spearman’s rho. It should be noted that GNSS was only correlated with Navigation and Age, and as expected, WQ subfactors correlated with each other.

Although the assumption of normality was not met for the different variables under study, the shape of the distribution was checked in terms of skewness and kurtosis. In this way, structural equation modeling was performed as well as PLS analysis. The proposed model was examined under structural equation modeling by employing the maximum likelihood method. An acceptable goodness of fit was found, as described in Figure 3. χ^2^/_degrees of freedom_ = 3.15; NFI = 0.844; IFI = 0.888; CFI = 0.887; RMSEA = 0.07.

The goodness of fit for the women group was found: χ^2^/_degrees of freedom_ = 2.19; NFI = 0.793; IFI = 0.876; CFI = 0.874; RMSEA = 0.08. The goodness of fit for the men group was found: χ^2^/_degrees of freedom_ = 2.09; NFI = 0.77; IFI = 0.866; CFI = 0.864; RMSEA = 0.08. With regard to the PLS approach, Figure 4 shows the inherent path for each group and the whole data set.

As shown in Table 3, a higher explained variance was reached for women in GPS scores rather than men. Moreover, a higher explained variance was found for navigation scores rather than GNSS ones.

## 4. Discussion

Global navigation satellite systems are a widely useable tool to obtain invaluable spatial–temporal data [27,28]. Research on self-perception in visuospatial navigation as well as attitudes towards the adoption of digital devices is of interest to understand possible barriers as well as the detrimental effects of digital devices [13]. In this way, the aim of this work was twofold, e.g., to propose a new questionnaire on attitudes towards global navigation satellite systems (GNSSs) and to evaluate a model and its differences according to gender and age. The new questionnaire proposal showed optimal psychometric properties. By correlating with the WQ Navigation subscale but not with the others, it showed content validity. Furthermore, the specific relation we found between the GNSS scale, and the Navigation factor is consistent with the idea that the egocentric strategy has several similarities with GNSS instructions. Indeed, a relation between the Distance Estimation factor—which is strictly related to the allocentric strategy—and the GNSS scale was not found, although the relation between the GNSS scale and Navigation was significant. Thus, the GNSS scale was able to discriminate between the navigational strategy and the psychological constraints produced by this process. Interestingly, an inverse relation between the GNSS scale and age was found, which could be considered a reflection of its usability for the older group, as described in previous literature [4,29]. This result points toward the need to further improve the usability of GNSSs in elderly individuals, whose topographical learning from an egocentric perspective has been found to be worse compared to younger individuals [7].

The theoretical model fit adequately, both for the whole data set and for the groups of men and women. Moreover, these results were replicated through the PLS approach, which was chosen to avoid biases that might occur because of the normality limitations of the data. An inverse relationship between age and technology adoption was found for women. Gender and age gaps have been commonly described in the field [30], and according to the current results, cohort effects might occur. Future lines of research should further examine the role of variables inherent in cognitive strategies with the current measures. Knowing that the psychometric aspects have been satisfactory, these results could support both direct and systematic replications in the future.

According to the SEM model, anxiety scores were a higher predictor for navigation in men, even if higher anxiety was found in women. Moreover, anxiety was a higher predictor for GNSS adoption in women, not supporting some of the previous results reported in the literature [19]. However, it should be noted that mixed results have been found in this regard. It should be noted that this effect might be moderated by age across groups, as suggested by the current results. On the other hand, distance estimation showed a similar relation with navigation for both groups. No relationship estimation of perceived GNSS use was found, and in turn, the latter variable does not seem to be correlated with navigation.

The main limitations that arise are that the sample was selected through non-probability sampling, which can introduce distortions in the results. Moreover, data were obtained through self-reporting. Thus, the measures have a strong subjective component relying on the individual’s perception. Further research work should examine self-perception with navigation strategies in a more objective way, such as measuring participants’ allocentric and egocentric strategies. Nevertheless, the navigation subscale of the Wayfinding Questionnaire is of interest as it has been shown to correlate with objective navigation abilities in both clinical and healthy populations [23,24], thus providing support to the present findings. Lastly, the study was conducted in Spanish (for the Spanish population). This issue limits generalizability to other populations, so psychometric adaptation studies are of interest.

In sum, the current results may be of interest on both theoretical and practical levels. At the theoretical level, they allow the application of classical models, such as UTAUT, to the adoption of this type of device [31], while at the practical level, they can promote the independence of the individual. The discovery of the detrimental effects of GNSSs on the human brain makes timelier than ever the construction of a unified framework for navigation tested in an interdisciplinary way. Even if alarming results have been found for GPS users, caution is advised here. First, the current approach is not only focused on losses through technology but also through gains and adoption opportunities [32]. Secondly, banning technological devices is the equivalent of denying digital language. Rather than censoring them, it is needed to live with them, enhancing their beneficial uses—for example, for elderly people. This includes understanding human cognition and technology display, as well as attitudes, as people of all groups deserve to be trained through various means to learn to live with technology. In the worst scenario where GNSSs really have a harmful effect on human cognition, compensatory mechanisms should be clarified.

## 5. Conclusions

The novelty of this work is its provision of a tool to assess attitudes toward global navigation satellite systems (GNSSs) across the lifespan and to study their role within a technological-adoption model. The main results can be summarized as follows: (i) the proposed scale of human spatial navigation attitudes towards global navigation satellite systems showed optimal psychometric properties; (ii) although statistically significant differences were found in WQ between men and women, these did not reach the level of statistical significance for the scores on attitudes towards global navigation satellite systems; (iii) similar results were found through the SEM and PLS approaches; (iv) by testing a model of human spatial navigation attitudes towards geopositioning technology, it was possible to discern a higher relationship with age in women.

## Figures and Tables

**Figure 1 sensors-22-03470-f001:**
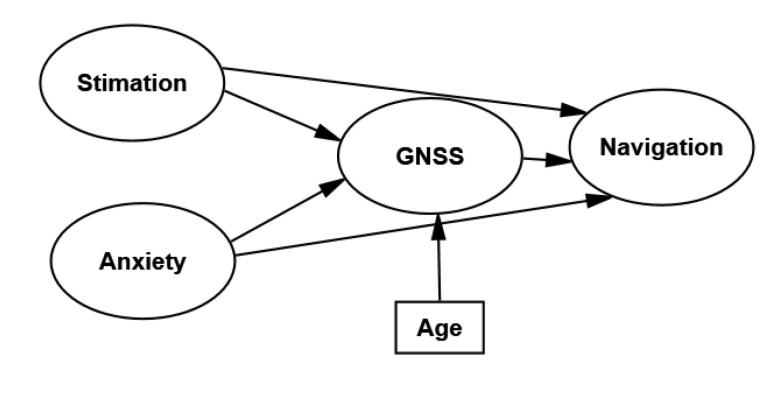
Proposed model on human spatial navigation attitudes towards global navigation satellite systems based on the UTAUT theory.

**Figure 2 sensors-22-03470-f002:**
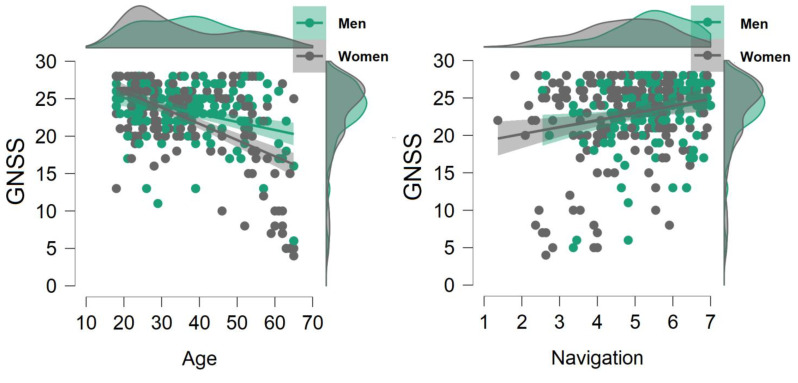
Spearman’s Partial Correlations for Age, Navigation, and GNSS, conditioned by gender.

**Figure 3 sensors-22-03470-f003:**
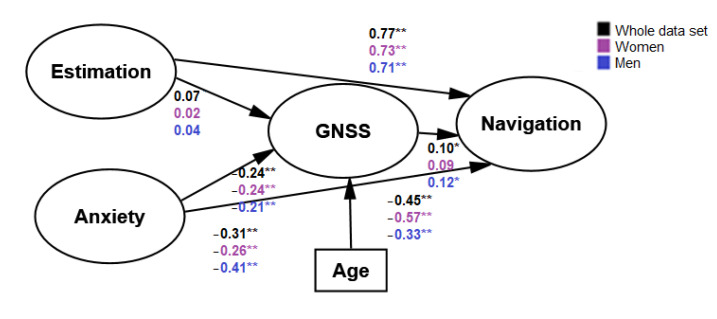
Estimation and loadings of the model proposed. ** = *p* < 0.001; * = *p* < 0.05.

**Figure 4 sensors-22-03470-f004:**
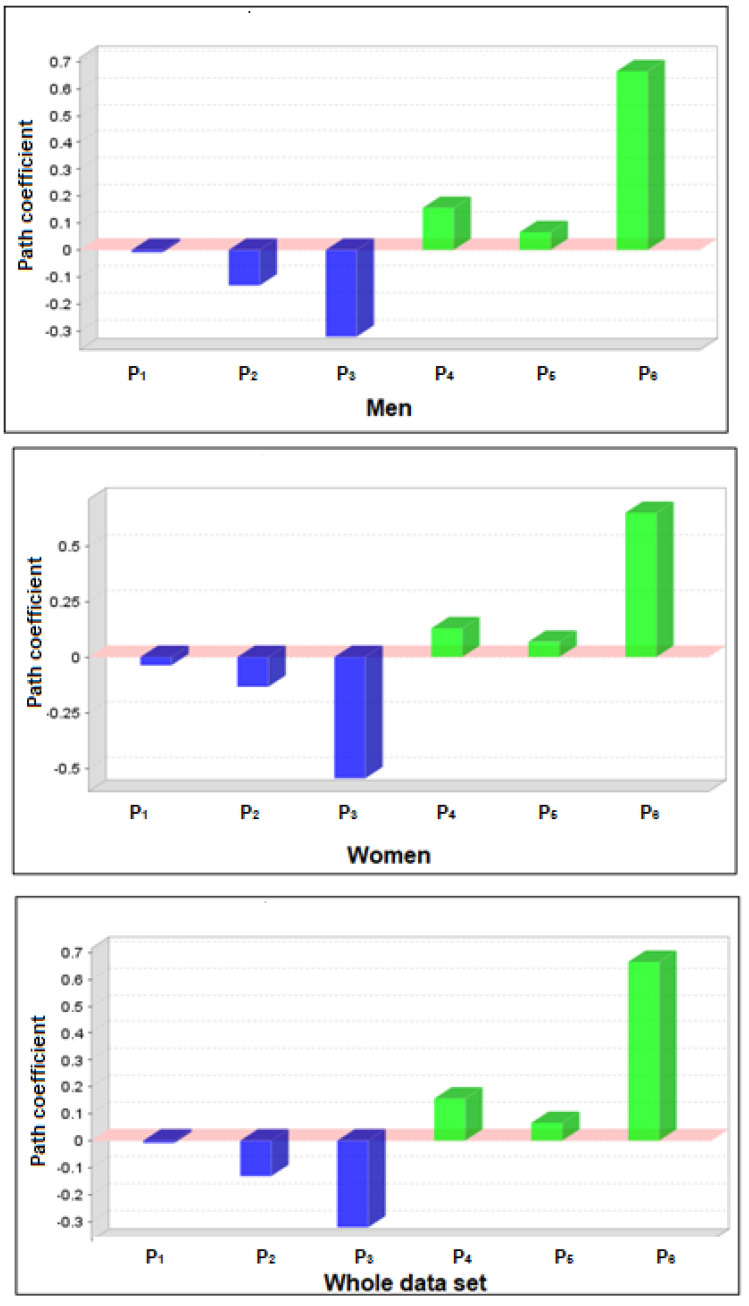
Path coefficients amongst variables for each gender group and the whole data set. P_1_ = Anxiety-GNSS; P_2_ = Anxiety-Navigation; P_3_ = Age-GNSS; P_4_ = GNSS-Navigation; P_5_ = Estimation-GNSS; P_6_ = Estimation-Navigation.

**Table 1 sensors-22-03470-t001:** Mann-Whitney U test employed to examine differences across gender, upper and lower CI, as well as the Rank-Biserial Correlation. Global navigation satellite systems = GNSS.

	W	*p*	Rank-Biserial	95% CI for Rank-Biserial Correlation	
				Lower	Upper
GNSS	14,819	0.771	0.006	−0.139	0.104
Navigation	484.00	<0.001	0.392	0.231	0.532
Estimation	4999.50	<0.001	0.438	0.283	0.570
Anxiety	2001.50	<0.001	−0.424	−0.559	−0.268

**Table 2 sensors-22-03470-t002:** Spearman’s Partial Correlations between AGE, WQ subfactors, and GNSS, conditioned by gender. ** = *p* < 0.001; * = *p* < 0.05.

Variable	Age	GPS	Navigation	Estimation
Age	-			
GNSS	−0.466 **	-		
Navigation	−0.099	0.201 **	-	-
Estimation	−0.012	0.082	0.682 **	-
Anxiety	0.078	0.069	−0.226 **	−0.128 *

**Table 3 sensors-22-03470-t003:** Error measures on the models under study, Q^2^ and R^2^.

		RMSE	MAE	MAPE	Q²	R^2^
Men	GNSS	466.014	358.632	23.466	0.280	0.10
	Navigation	85.848	68.900	17.382	0.471	0.52
Women	GNSS	466.444	357.992	23.522	0.281	0.30
	Navigation	85.986	69.162	17.464	0.469	0.49
Whole data set	GNSS	443.756	335.550	21.409	0.208	0.22
	Navigation	78.783	62.721	14.698	0.534	0.55

## Data Availability

Data are available upon request.

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
