# Peer review of "Testing a Model of Human Spatial Navigation Attitudes towards Global Navigation Satellite Systems"

_sensors, 2022, doi:10.3390/s22093470_

Round 1

Reviewer 1 Report

The authors present a study on the barriers for adoption for “GPS” with respect to variable such as age and gender. 

The paper is reasonably well written with respect to fluent English usage, although there are a few areas that could be cleaned up. For instance, the introduction is jumbled and does not compel the reader.

The paper requires a few areas to be addressed before acceptance.

The abstract should be rewritten to be more compelling.

The term “GPS” refers to the U.S. instantiation of a “global navigation satellite system”, which is the general term.  I suggest using this term unless these results are specific to GPS usage.  Other systems such as Google Maps may not use GPS in denied environments.  I think the results can be applied to a more general way-finding/mapping system, and that should be stated.  The term “positioning device” was used in the survey presented in the Appendix.

Line 40: There is a really critical statement/hypothesis on “habitual use of GPS…” which I think is a strong statement that is the driver for this study, yet it is buried.  I would highlight it more.

Line 66: It is interesting the hear the statement that “women will not adopt digital technology to geo-position themselves”.  Are you sure this is the outcome of the study presented.  While there are numerous studies that indicate gender difference in geospatial and spatial abilities, it would seem a GPS system, per your hypothesis, actually limits the need to understand “geospatialness.”

Line 101: what is meant by “visuospatial navigation complaints”?

Paragraph 110: Were all participants Spanish and did this have any bearing on the results?  How might the results be different for other nationalities and cultural groups?

Author Response

The authors present a study on the barriers for adoption for “GPS” with respect to variable such as age and gender. 

The paper is reasonably well written with respect to fluent English usage, although there are a few areas that could be cleaned up. For instance, the introduction is jumbled and does not compel the reader.

Thank you for your comments and invaluable time and help. Reformulations are highlighted in yellow.

The paper requires a few areas to be addressed before acceptance.

The abstract should be rewritten to be more compelling.

We have reformulated it as suggested.

The term “GPS” refers to the U.S. instantiation of a “global navigation satellite system”, which is the general term.  I suggest using this term unless these results are specific to GPS usage.  Other systems such as Google Maps may not use GPS in denied environments.  I think the results can be applied to a more general way-finding/mapping system, and that should be stated.  The term “positioning device” was used in the survey presented in the Appendix.

Thank you for this valuable point. We have reformulated it as suggested.

Line 40: There is a really critical statement/hypothesis on “habitual use of GPS…” which I think is a strong statement that is the driver for this study, yet it is buried.  I would highlight it more.

We agree with the reviewer and we have reformulated it.

Line 66: It is interesting the hear the statement that “women will not adopt digital technology to geo-position themselves”.  Are you sure this is the outcome of the study presented.  While there are numerous studies that indicate gender difference in geospatial and spatial abilities, it would seem a GPS system, per your hypothesis, actually limits the need to understand “geospatialness.”

Thank you for this approach. With your permission, we would like to include it in the discussion with more nuances on it.

Line 101: what is meant by “visuospatial navigation complaints”?

It is related to one of the subscales in WQ. We have reformulated it to make it clear this point.

Paragraph 110: Were all participants Spanish and did this have any bearing on the results?  How might the results be different for other nationalities and cultural groups?

We have marked this as a limitation of the study and a future line of research. Thank you for the suggestion and your invaluable comments.

Reviewer 2 Report

The paper deals with the evaluation of the social aspects of geo-positioning systems and technologies. 

 The language used in the manuscript is not technical and some claims are not quite correct. For example "optimal route will be suggested by the GPS" can be considered to be a false statement as GPS can only provide position estimates. The estimation of the optimal route requires some wayfinding algorithm and additional information (i.e. available routes on a map, information about traffic, etc.). 

Moreover, for example, GPS in the abstract seems to represent geo-positioning system and technology, however, in the text of the manuscript it seems to represent Global Positioning System.

It is unclear how data analysed in the study were collected. The presented questionnaire has only  4 questions, therefore, it is obvious that not all required data could be collected solely by the presented questionnaire. 

The authors do mention the use of the Wayfinding Questionnaire, however, such questionnaire cannot be found in provided references 18 and 19. 

Author Response

The paper deals with the evaluation of the social aspects of geo-positioning systems and technologies.

Thank you for your comments and invaluable time and help. Reformulations are highlighted in yellow.

 The language used in the manuscript is not technical and some claims are not quite correct. For example "optimal route will be suggested by the GPS" can be considered to be a false statement as GPS can only provide position estimates. The estimation of the optimal route requires some wayfinding algorithm and additional information (i.e. available routes on a map, information about traffic, etc.).

Thank you for the remark. We have reviewed the manuscript carefully, paying special attention to these issues.

Moreover, for example, GPS in the abstract seems to represent geo-positioning system and technology, however, in the text of the manuscript it seems to represent Global Positioning System.

Thank you once again, this is an issue also highlighted by reviewer one, reformulated as suggested.

It is unclear how data analysed in the study were collected. The presented questionnaire has only  4 questions, therefore, it is obvious that not all required data could be collected solely by the presented questionnaire.

Two questionnaires were used. The first one was created for the purpose of measuring attitudes towards GPS. Its scoring follows psychometric recommendations, creating a single measure composed of four items (GNSS measure). On the other hand, the WQ questionnaire was used, which is a reference in the literature and measures three factors: navigation, estimation and anxiety. We have reformulated it to make it clear

The authors do mention the use of the Wayfinding Questionnaire, however, such questionnaire cannot be found in provided references 18 and 19.

We have included it in the appendix

Reviewer 3 Report

Please, apply the revisions done in the attached file, including "Coefficients of Paths" in the figures.

Author Response

Thank you for your invaluable time and help to improve the current manuscript.

Round 2

Reviewer 2 Report

Authors should clearly state what is the novelty and contribution of the paper in the introduction of the manuscript. 

The English language has to be improved, there are a handful of grammatical mistakes manuscript. 

Questions 6 and 22 in the WQ questionnaire are not translated into English, while others are. I would suggest translating all the questions in order to make the questionnaire readable to a wider audience. 

Is there any difference between "GNSS-scale" and "GPS-scale"? Both terms are used in the manuscript, which is a bit confusing as Global Positioning System (GPS) is just one of the Global Navigation Satellite Systems (GNSS), other representatives of GNSS can be for example GLONASS, Beidou and Galileo. 

The quality of figures is very poor, I would recommend using figures in vector format where possible or in lossless format (e.g. tiff or png).

Author Response

Dear reviewer, thank you for your invaluable time and help.

Please, find new changes in blue.

Authors should clearly state what is the novelty and contribution of the paper in the introduction of the manuscript. 

We have extended the las sections to addess this issue.

The English language has to be improved, there are a handful of grammatical mistakes manuscript. 

An english native speaker has reviewed the manuscript.

Questions 6 and 22 in the WQ questionnaire are not translated into English, while others are. I would suggest translating all the questions in order to make the questionnaire readable to a wider audience. 

We are really sorry for this mistake.

Is there any difference between "GNSS-scale" and "GPS-scale"? Both terms are used in the manuscript, which is a bit confusing as Global Positioning System (GPS) is just one of the Global Navigation Satellite Systems (GNSS), other representatives of GNSS can be for example GLONASS, Beidou and Galileo. 

It should be GNSS. We are sorry for the mistake.

The quality of figures is very poor, I would recommend using figures in vector format where possible or in lossless format (e.g. tiff or png).

We have reformulated figures as suggested.

Round 3

Reviewer 2 Report

Abbreviations in the text are not well used.

For example,  line 35 is when GPS is used mentioned, however, the first definition of GPS abbreviation can be found in line 84.

With GNSS it is even worse. The abbreviation should be defined when the name is first used and can be used in all the text after. Since GNSS is defined it is not necessary to use the full form. 

The novelty of the paper is questionable, GNSS is a technology that has been around since the 1980s and can now be considered to be widely available and widely adopted technology.  

Author Response

Dear reviewer,

We agree that some terms should be better addressed in the previous version. We tried to make a better of revision as suggested. We have only kept the GPS term in line 87-89, as the manuscript cited employed this term. We really appreciate the remark. GNSS is defined in its first use now.

Lastly, we also agree that GNSS is a technology studies for decades. We have changed both introduction and discusion to address the interest in older adults groups, as differences might occur here. 

We hope this new version is satisfactory. In any case, we really appreciate all your invaluable time and help to improve the current manuscript.